# scikit-learn Pipelines meet Knowledge Graphs
## The Python kgextension Package

Tabea-Clara Bucher, Xuehui Jiang, Ole Meyer, Stephan Waitz,
Sven Hertling[0000−0003−0333−5888], and Heiko Paulheim[0000−0003−4386−8195]

Data and Web Science Group, University of Mannheim, Germany
{tbucher,xujiang,olmeyer,swaitz}@mail.uni-mannheim.de,
{heiko,sven}@informatik.uni-mannheim.de

**Abstract.** Python is currently the most used platform for data science and machine learning. At the same time, public knowledge graphs have been identified as a valuable source of background knowledge in many data science tasks. In this paper, we introduce the kgextension package for Python, which allows for using knowledge graph in data science pipelines built in Python. The demo shows how data from public knowledge graphs such as DBpedia and Wikidata can be used in data mining pipelines based on the popular Python package scikit-learn. We demonstrate the package's utility by showing that the prediction accuracy on a popular Kaggle task can be significantly increased by using background knoweldge from DBpedia.

**Keywords:** Python · scikit-learn · Knowledge Graph · Background Knowledge · Data Mining

## 1  Introduction

According to a recent poll, Python is the most used platform for data science and machine learning.[1] At the same time, public knowledge graphs have been acknowledged as a valuable source for background knowledge in such tasks [14]. While packages such as rdflib[2] are quite popular for processing knowledge graphs, they do not build a bridge between graph processing and widely used data mining packages, such as scikit-learn[3].

In this paper, we present the kgextension package for Python[4], which builds exactly that bridge. It builds on the ideas of previous implementations for *Weka* [7] and *RapidMiner* [10]. The package provides functionalities for linking a dataset to public knowledge graphs, as well as for extracting features from those graphs. It comes with preconfigured connections to DBpedia and Wikidata, but can also be used with custom SPARQL endpoints and local RDF dumps.

---

[1] https://www.kdnuggets.com/2019/05/poll-top-data-science-machine-learning-platforms.html
[2] https://github.com/RDFLib/rdflib
[3] https://scikit-learn.org/
[4] https://github.com/om-hb/kgextension

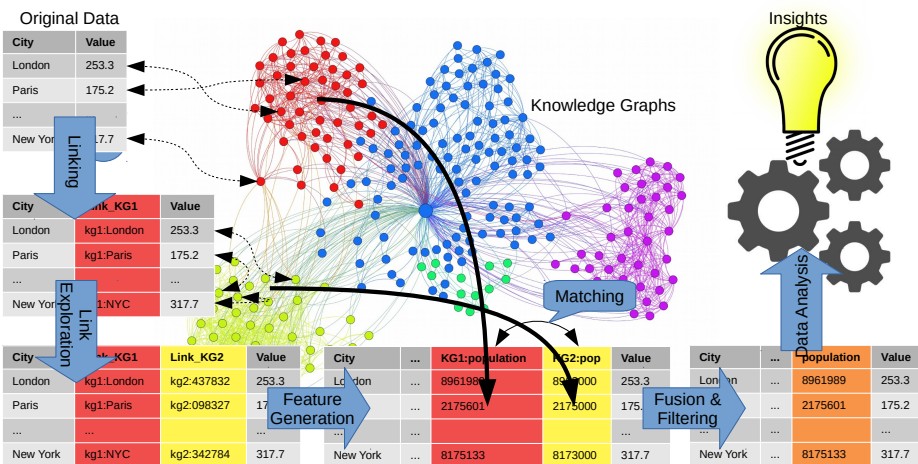

Fig. 1: Data analysis pipeline using background knowledge from knowledge graphs

## 2    Package Functionalities

A data analytics pipeline using background knowledge from knowledge graphs typically comprises different steps, as shown in Fig. 1. The final step is performing the actual data analysis, for which built-in methods of `scikit-learn` or other data mining packages are used. The remaining steps are supported by `kgextension`.

### 2.1    Linking and Link Exploration

The first step is to identify entities from the dataset to analyze in a knowledge graph. For example, on a dataset of cities, this step would be in identifying the corresponding cities in a knowledge graph. To that end, different entity linkers are available, which implement techniques such as user-defined URI patterns[5], lookup via SPARQL queries, or wrappers for specific services such as DBpedia Lookup[6]. Once links to one knowledge graph are established, links to other datasets (e.g., `owl:sameAs`) may be explored for generating additional links.

### 2.2    Feature Generation

In the next step, features are extracted from the linked entities in the knowledge graph. The package implements a number of techniques, ranging from the creation of individual features for datatype properties (e.g., the city population),

---

[5] such as `http://dbpedia.org/resource/*ENTITY*`

[6] `https://lookup.dbpedia.org/`

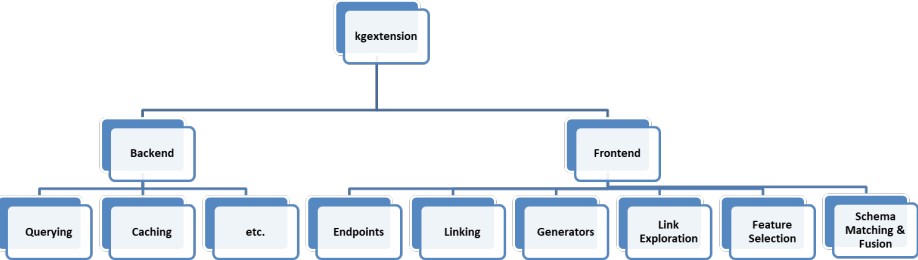

Fig. 2: Package Structure

binary features for types (e.g., binary features for types such as `capital city`, `european city`, etc.), and different flavours of aggregation of properties [11] (e.g., using TF-IDF based measures). Moreover, custom SPARQL queries can be used for constructing specific features.

### 2.3  Feature Filtering

While `scikit-learn` provides a lot of generic techniques for feature filtering[7], the `kgextension` package also implements a number of specific methods from the literature, which consider the ontology underlying the knowledge graph for guiding the feature selection process. These methods do not only use internal measures such as information gain, but also take, e.g., the hierarchy in the ontologies into account for identifying the most distinctive features [5,6,12,19].

### 2.4  Matching and Fusion

When extracting features from more than one knowledge graph, there might be duplicate attributes (e.g., population values extracted from Wikidata and DBpedia). The `kgextension` includes a set of methods for identifying similar attributes (e.g., based on string similarity of the attribute names, or on value overlap), and includes a number of heuristics for fusing the values of joined attributes (e.g., voting, averaging, etc.).

### 2.5  Other Functionalities

As shown in Fig. 2, the `kgextension` package also comprises a number of useful backend functionalities, e.g., for efficient access to endpoints and caching. They facilitate an efficient execution of the overall pipeline.

---

[7] `https://scikit-learn.org/stable/modules/feature_selection.html`

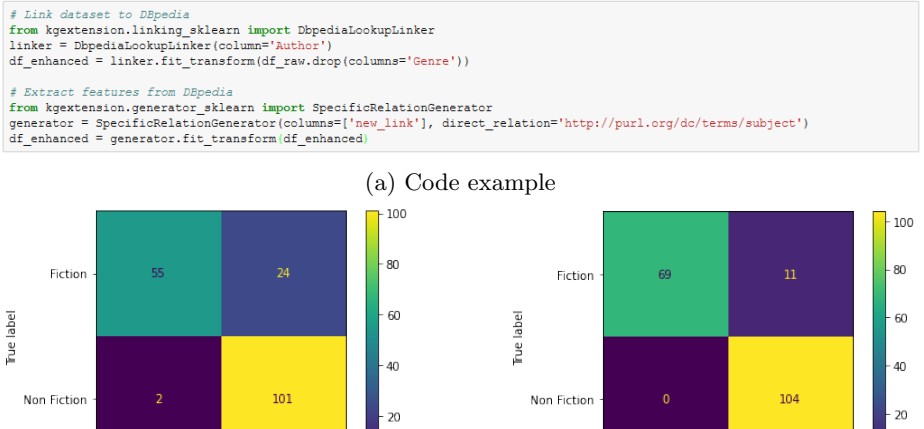

```
# Link dataset to DBpedia
from kgextension.linking_sklearn import DbpediaLookupLinker
linker = DbpediaLookupLinker(column='Author')
df_enhanced = linker.fit_transform(df_raw.drop(columns='Genre'))

# Extract features from DBpedia
from kgextension.generator_sklearn import SpecificRelationGenerator
generator = SpecificRelationGenerator(columns=['new_link'], direct_relation='http://purl.org/dc/terms/subject')
df_enhanced = generator.fit_transform(df_enhanced)
```

(a) Code example

(b) Confusion matrix without added features

(c) Confusion matrix with features added from DBpedia

Fig. 3: Adding features from DBpedia significantly improves the results

## 3 Demonstration Contents

We demonstrate an end to end use case (i.e., from a dataset to a prediction), which is also available online as a Jupyter notebook[8]. In this use case, we use a prediction task from Kaggle[9] and show how to extend the dataset with information from different public knowledge graphs using the different functionalities of the package. The prediction target is to classify books in fiction and non-fiction books. We show that by using a few simple Python commands, the performance increases significantly from an accuracy of 0.86 to an accuracy of 0.94, and the number of wrongly classified examples reduced to more than half, as shown in Fig. 3. Since the notebook is interactive, different variants can be explored together with attendees of the demo.

## 4 Future Developments

The kgextension package itself is developed in a modular fashion, which allows for integrating new functionalities. Thus, for the future, we are planning to integrate, e.g., novel methods for linking [2]. Since knowledge graph embeddings have been proven useful for many data science tasks [1,13,15], we also plan to integrate libraries for creating knowledge graph embedding vectors [16], as

---

[8] https://github.com/om-hb/kgextension/blob/master/examples/book_genre_prediction.ipynb

[9] https://www.kaggle.com/sootersaalu/amazon-top-50-bestselling-books-2009-2019

well as adapters for repositories for pretrained knowledge graph embeddings [9]. Moreover, on the knowledge graph access side, we plan to integrate efficient generators for Triple Pattern Fragment endpoints [17] and HDT files [3].

Moreover, the framework provide an interesting test bed for designing comparative studies of different public knowledge graphs [4] on various downstream tasks [8], a field which has not been much considered yet [18].

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
