# OpenReview forum: "scikit-learn Pipelines meet Knowledge Graphs - The Python kgextension Package"
_eswc-conferences.org/ESWC/2021/Conference/Poster_and_Demo_Track — ESWC2021 P&D_

### Official Review · AnonReviewer4 · 2021-04-12
**Low novelty but high utility; well written and presented**

**Rating:** 7
**Confidence:** 4

**Review:**

This paper presents a new Python package for performing tasks over Knowledge Graphs (KG). Among these tasks, this package provides:

- a wrapper for entity linking by using DBpedia Lookups, DBpedia Spotlight, or using custom SPARQL queries
- for specified KG entities, a way to retrieve KG information taking as reference their neighbors
- a wrapper for local SPARQL endpoints and remote ones to Wikidata and dbpedia
- given a list of entities, this tool retrieves for each of them a list of same_as entities recursively.
- implements scikit-learn interfaces to interact in scikit-learn pipeline

This paper is clear and well written.


Although most of the functionalities are not novel solutions ( due to it is composed of wrappers of existing libraries such as DBpedia Lookups, DBpedia Spotlight, SPARQLWrapper, and rdflib), this library facilitates a lot of work for related tasks, and it is worth it to be presented. It is well documented, uploaded to a recognized python repository (PyPi), and works fine according to some simple tests that I conducted.

In the paper, I would have liked to see some detail about its performance processing high volumes of data, for example, all DBpedia. In my experience, Pandas has a slow performance for a large volume of data.


**Anonymity:**

Yes, I would like my review to remain anonymous.

---

### Official Review · AnonReviewer1 · 2021-04-13
**Interesting demo, could be useful to broader community**

**Rating:** 8
**Confidence:** 4

**Review:**

present the kgextension package for Python, which builds a bridge between graph processing and widely used
data mining packages, such as scikit-learn. The authors will demonstrate an end to end use case (i.e., from a dataset to a prediction),
which is also available online as a Jupyter notebook.

Although I haven't tried out the notebook myself, the tool sounds very promising and could end up being widely used if the authors plan the demonstration right. Hence, I recommend acceptance.

**Anonymity:**

Yes, I would like my review to remain anonymous.

---

### Official Review · AnonReviewer2 · 2021-04-13
**A "facilitator" for KG research**

**Rating:** 8
**Confidence:** 4

**Review:**

The paper introduces kgextension, which make available a set of utilities for link exploration and feature generation.

The paper is clear, the code available and well documented.  There is no evaluation in the paper, but a jupyter notebook in the repository (clearly linked in the paper) serves to the scope.

The library is a good contribution to the community, which can more easily tackle classic KG-related tasks.

Minor critics: it is not clear from the paper (nor from the repository) which part of the library are coded from scratch, and which others are imports and harmonisations of other libraries.

**Anonymity:**

Yes, I would like my review to remain anonymous.

---

### Official Review · AnonReviewer3 · 2021-04-14
**The excellent work to bridge knowledge graph and machine learning**

**Rating:** 9
**Confidence:** 3

**Review:**

The paper describes the Python package to import and process knowledge graphs data like DBpeida and Wikidata in order to use them for machine learning. It is simply useful for users in both the knowledge graph community and the machine learning community.   The paper successfully explained the functions of the package in a concise way and showed the demonstration. The source code and the examples are also available on Github.

**Anonymity:**

Yes, I would like my review to remain anonymous.

---

### Decision · Program_Chairs · 2021-04-19

Accept